# Strategy for Improving the Indoor Environment of Office Spaces in Subtropical Cities

**Wen-Pei Sung [1,\*], Ting-Yu Chen [1] and Chun-Hao Liu [2]**

1. Department of Landscape Architecture, Integrated Research Center for Green Technologies, National Chin-Yi University of Technology, Taichung 41170, Taiwan; tychen@ncut.edu.tw
2. Department of Landscape Architecture, National Chin-Yi University of Technology, Taichung 41170, Taiwan; 4a734002@gm.student.ncut.edu.tw
* Correspondence: drwpsung@gmail.com or wps@ncut.edu.tw; Tel.: +886-4-2392-4505#8112; Fax: +886-4-2393-0737

**Abstract:** Taiwan is located in a subtropical region with high temperatures and humidity. Office spaces are located in air-enclosed rooms in buildings, where doors and windows remain closed and only a central air-conditioning system provides temperature adjustment and ventilation. Most offices in this area have office seating areas, document storage areas on both sides of the office, and multi-function devices, which can cause sick office syndrome in the employees. This study applied environmental monitoring technology to analyze the architectural form and indoor and outdoor air quality to propose improvement strategies addressing indoor temperature, relative humidity and air quality. Quality indices were used created to evaluate the improvement efficiency. The analysis results showed that the indoor temperature and relative humidity in staff seating areas can be effectively improved. The statistical analysis results of improved efficiency for $PM_{2.5}$, $PM_{10}$ concentrations and total suspended particulates showed that the average indicator values have been raised from 0.05 to 1.5, 2.45 to 4.02 and 0.91 to 3.54, respectively, for staff seating area and $-0.01$ to 2.82, 0.15 to 3.91 and 1.25 to 7.25, respectively, for photocopier areas. The ambient air quality of this office space has been significantly improved. This study can serve as an example of air quality improvement in traditional common office spaces.

**Keywords:** office space; air quality; suspended particulates; central air conditioning; external air exchange





## 1. Introduction

There are two main substances of very high concern of air quality in the office. One is volatile organic substances, commonly found in wall paints such as paint, or furniture corrosion or pesticides like formaldehyde. The other category is particulate matter in the air, and common detections include $PM_{2.5}$ and $PM_{10}$, which are the source of outdoor air pollution inflow [1,2] or are deteriorated according to the poor ventilation of the parking lot with its high airtightness [3]. In the case of an enclosed office building, it was accumulated dust and related business electrical equipment, such as photocopiers. There are some features, such as year-round closed windows, neat office automation style office grids, central air-conditioning systems and business electrical equipment in office spaces at main cities of Taiwan. Common sources of pollution in offices include paint, furniture, office machines and even long-term under cleaning air-conditioning filters.

There have been many studies shown that air quality affects the attention of people who use it inside the space, which showed that poor indoor air quality can lead to a decline in academic performance and affect students' comfort and attention, which can affect their ability to learn in class [4–8]. The literature also points out that intelligent window (SWV) systems and appropriate air-exchange devices can effectively improve indoor air quality and thermal comfort [9–11].

The shape of the opening of a central air conditioner has a great influence on the air exchange efficiency. The Center for Building Performance and Diagnostics (CBPD) at Carnegie Mellon University [12] states that indoor air quality (IAQ) in the workplace is closely related to employee health, comfort and satisfaction. Especially, carbon dioxide and aerosols [13–15] in the air were identified as key factors in office staff satisfaction. Therefore, the literature recommends that the design and planning of projects, such as windows that can be opened and closed normally, dedicated exhaust devices, the air-conditioning one-way return system density, and medium- and low-separation height compartments, should ensure increased office comfort and satisfaction. Among the main factors in reducing air quality in offices are the sources of indoor air pollution [16–19]. An evaluation study [16] at an office building in Iran showed that the majority of employees found the dusty air unacceptable and suffered fatigue and headaches, and 21% felt that the air quality conditions in the working environment were unsuitable. Research results [20,21] indicated that cleaning, while beneficial, also poses risks and is the source of nearly 20% of the indoor pollution.

Since indoor air quality is a major problem in human health, the use of improper cleaning products can lead to a unique formaldehyde release power curve, showing that cleaning is a source of formaldehyde pollution [22]. In fact, the use of essential oil-based cleaning products [20,21] results in long-term increases in indoor formaldehyde concentrations due to the sustained release of high concentrations for several hours after cleaning. Another study [23] also reported that viruses in the air could easily mix with air pollutants, especially in crowded and poorly ventilated environments, increasing the likelihood of transmission. Therefore, air quality appears to have been a key environmental factor in the coronavirus disease (COVID-19) pandemic [24]. The accumulation of indoor air pollutants seems to be closely related to sick building syndrome [25]. Therefore, maintaining good air quality will allow the occupants of a building to maintain good health.

A traditional urban office in central Taiwan was investigated in this study. Such offices feature confined spaces, central air-conditioning systems, high space utilization, combined clerical seating and electrical equipment areas, heavy carpeting on the floor, and the division of the overall space by seat partitions and bookcases, which reduce circulation. After on-site investigation and considering financial issues, the most economical strategy was chosen for amelioration of the indoor environment of this office: 1. add a mandatory ventilation system; 2. restore the air exchange system; 3. clean the air conditioning filter; 4. add ceiling openings for the installation of multiple air-back filters; 5. replace the impervious office grid; and 6. replace the heavy carpets; which only cost ten thousand U.S. dollars, were proposed to modify the indoor environment and air quality of this test office. Then, on-site experiments were conducted to compare the real improved benefits before and after changes. To evaluate the improved efficiency, the environmental improvement performance indicator [26,27] and the indicator of smaller-the-better process capability [28,29] were adapted to evaluate the changes in indoor temperature and humidity before and after the implementation of improvement plans in this study. The evaluation method of quality condition [30] and the fraction defective analysis would be used to quickly evaluate the quality condition for improving efficiency of indoor air quality before and after the implementation of the improvement plans.

## 2. Experimental Setting

Taichung City is located in central Taiwan, and the experiment setting was situated in the central part of Taichung City. The city is surrounded by hills and sits in a basin, as shown in Figure 1. The region has a subtropical monsoon climate with a strong radiation cooling effect, and Taichung is one of the few cities in Taiwan with annual low temperatures averaging close to 20 °C. However, being located in a basin, Taichung has an obvious heat island effect, with high temperatures exceeding 35 °C in summer and relative humidity often as high as 80%. Due to the limited upward flow of the atmosphere, which can easily

lead to air pollution, most office buildings are designed to be closed spaces wherein the temperature and humidity are adjusted through the use of central air conditioning systems.

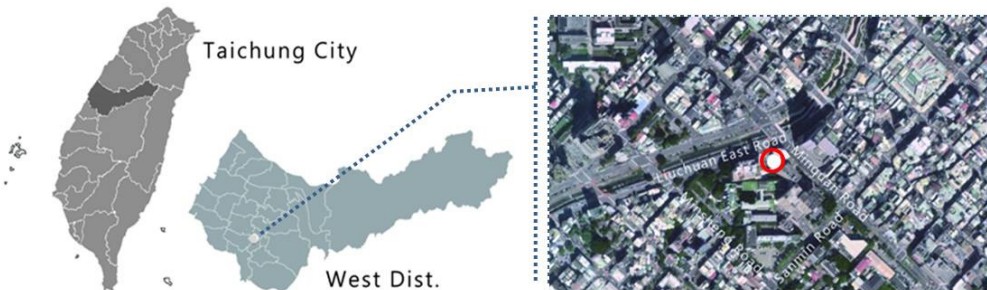

**Figure 1.** The relative location of this experimental field in Taichung, Taiwan.

## 2.1. Description of Test Office

This study mainly focused on the interior space of an office building (Figure 2) in the Taichung urban area, where experiments were performed and then improvement plans were implemented. Field tests were conducted before and after the implementation of the improvement plan to verify the changes. The office space of this field test was 1000 square meters, divided into three sections. The center held 38 office desks separated by partitions. Document storage areas occupied both sides of the office space, and the electrical equipment area was located in the front of the office. The space had two entrances and exits. Along the side of the office facing the road was a row of windows. Due to the exhaust from the passing vehicles, the air quality by the road was poor, so the windows on this side usually remained closed. Central air conditioning was applied for ventilation in summer, while in winter, the front and rear doors were opened and fans were used for ventilation. The electrical equipment area in the front of the office had two large photocopiers for printing, filing cabinets, and binders. Along both sides and the back of the office were filing cabinets for reference materials and other documents required by the office staff.

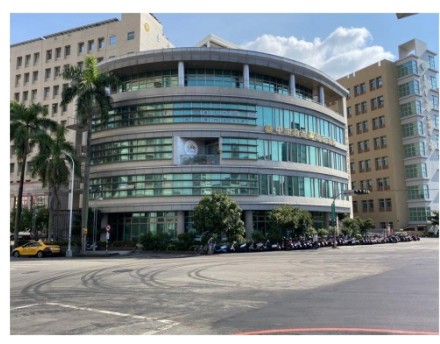 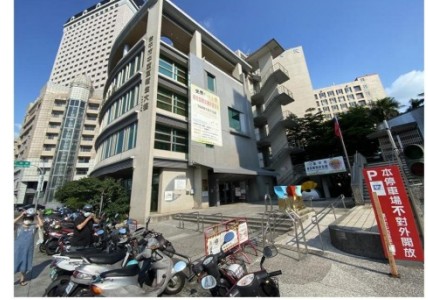

(**a**) Front photo of this test site.      (**b**) Office entrance of this test site.

**Figure 2.** The test site of office building in urban area.

## 2.2. Experimental Equipment

The equipment used in this experiment for the suspended particulate analysis was a multi-functional dust meter from Metropolis Instruments Co., Ltd. in the United States, model AEROCET 531S. The meter uses a laser dipole (780 nm) to measure $PM_1$, $PM_{2.5}$, $PM_4$, $PM_7$, $PM_{10}$, Total Suspended particles (TSP) and other items and their concentrations simultaneously. Particle sizes of 0.3 μm, 0.5 μm, 1 μm, 5 μm, and 10 μm can be measured simultaneously, up to a maximum of 3,000,000 particles. The maximum measurable mass concentration is 1000 μg/m$^3$ at a flow rate of 2.83 L/min. The temperature and humidity in the test space were measured with a comprehensive indoor environmental quality tester

from the German company METREL, model MI6401. This device can also be used for thermal comfort measurements, including air temperature, wind speed, relative humidity/dew point, and illumination, as well as black sphere temperature and contact temperature. It is suitable for measuring temperatures of $-20$ to $60\ ^\circ\text{C}$ and humidity of up to 100% RH. The air flow rate can be adjusted to 0.05–9.99 m/s and 10.0–20.0 m/s

This study first explored the actual experiences of employees by conducting field interviews of the office workers. The results indicated that they felt uncomfortable working long hours in the indoor environment due to the poor air quality in the confined space, and as many as 86% of the employees had developed sick building syndrome. Therefore, a comprehensive indoor environmental quality tester was installed in the middle of the office to measure changes in the overall temperature and humidity and to analyze the air quality. The measured points in this test office were shown in Figure 3, three of the indoor multi-functional dust meters were setup at this test office, two at the staff seating area (point B) and one at the photocopier point (point A), shown in Figure 4, to measure variations in air quality, under the assumption that this area was likely to have high particle concentrations and was a source of the pollution. The multi-functional dust gauges were installed at these three positions to detect variations in particulate matter.

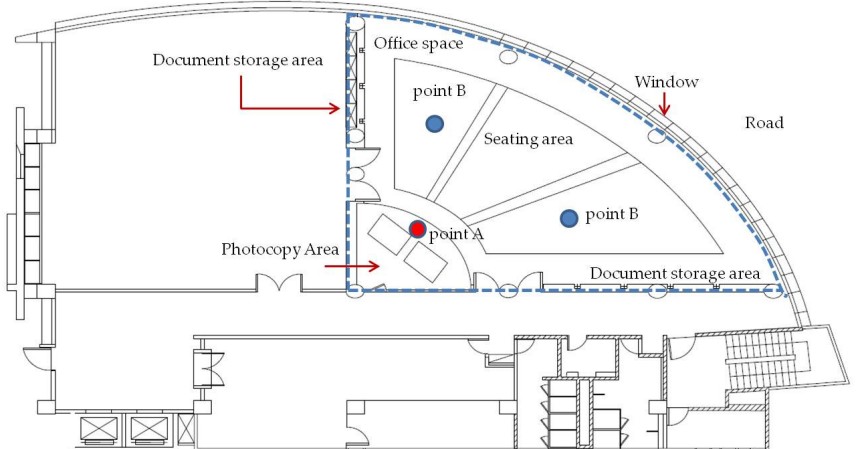

**Figure 3.** The measured points of the test office.

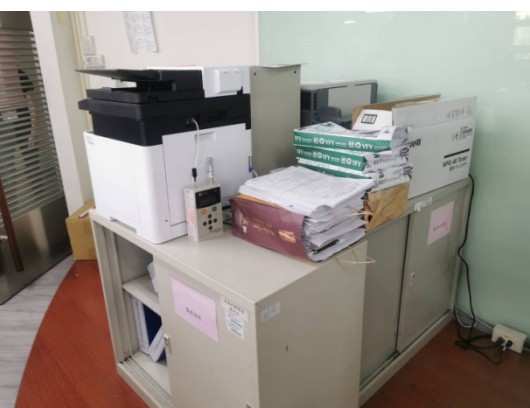

**Figure 4.** Detected device installed at photocopy area.

## 3. Methodology

To assess and verify the substantial benefits of the proposed improvement plans, process capability evaluation methods were employed. These cited evaluation modes are described as follows:

### 3.1. The Evaluation Indicator of Improving Efficiency of Indoor Temperature and Relative Humidity

Economical improvement strategies were proposed to improve the indoor environment in this study based on the characteristics of the study site. To evaluate and verify the efficiency of the proposed strategies, the environmental improvement performance indicator [26,27] was employed in this study to evaluate the indoor temperature and humidity before and after the implementation of the improvement plan. This index is widely used to evaluate improvements in manufacturing efficiency in industry. Therefore, this index was used to assess the improvements in environmental temperature and humidity. The environmental improvement performance indicator is expressed as follows:

$$C_i = \frac{(\mu - T)^2}{d^2} + \frac{\sigma^2}{d^2} \tag{1}$$

where:

$\mu$ is process mean;

$T$ is the target value of indoor temperature or humidity;

$d$ is the half of different between set upper value and set bottom value of indoor temperature or humidity;

$\sigma$ is process deviation.

Actually, when the value of $C_i$ is smaller, both of the $\frac{(\mu - T)^2}{d^2}$ and $\frac{\sigma^2}{d^2}$ are smaller as well. Smaller values indicate that the indoor temperature and humidity are close to the target values. Larger values of $C_i$ mean that the improvement strategy has not achieved the expected improvement efficiency. The tolerance ranges of temperature and relative humidity for people in Taiwan are different from those of people in other countries. Therefore, an indoor temperature range of 24.2–29.5 °C and a relative humidity range of 45–60% [31] were used as the target values in this study. These results are suitable for the residents, living in Taiwan.

### 3.2. The Evaluation Indicator of Indoor Air Quality

In this study, the $PM_{2.5}$ and $PM_{10}$ concentrations and the total suspended particles (TSP) were measured before and after the implementation of the improvement plan to evaluate the change in indoor air quality. Lower $PM_{2.5}$ and $PM_{10}$ concentrations and TSP indicate better indoor air quality. Therefore, an indicator of the-smaller-the-better process capability [28,29] was used before and after the implementation of the improvement plan to evaluate the efficiency of the plan. The $C_{pu}$ measure of the-smaller-the-better process capability is expressed as follows:

$$C_{pu} = \frac{USL - \mu}{3\sigma} \tag{2}$$

where:

$USL$ is the upper specification limit;

$\mu$ is process mean;

$\sigma$ is process deviation.

This index is useful for evaluating performance before and after an improvement. To quickly indicate the improvement efficiency, five quality conditions proposed by Pearn and Chen [30] were employed to evaluate the improvement efficiency. The advantage of this indicator is that it indicates improvements in performance based on the value of the indicator relative to the data on the five quality conditions, listed in Table 1. Table 1 shows that a process is "inadequate" if the value of $C_{pu}$ is less than 1.00. If so, the process cannot achieve the predetermined improvement efficiency. A process is called "capable" if the value of $C_{pu}$ ranges from 1.00 to 1.33; such a value indicates that the proposed improvement strategy needs to be modified. A process is called "satisfactory" if the value of $C_{pu}$ ranges from 1.33 to 1.50; it shows that the improvement strategy is only satisfactory. A process is

termed "excellent" if the value of $C_{pu}$ is between 1.50 and 2.00. If the value of $C_{pu}$ exceeds 2.00, a process is considered "super".

**Table 1.** The five quality conditions.

| Quality Condition | $C_{pu}$ Values |
|---|---|
| Inadequate | $C_{pu} < 1.000$ |
| Capable | $1.000 \leq C_{pu} < 1.333$ |
| Satisfactory | $1.333 \leq C_{pu} < 1.500$ |
| Excellent | $1.500 \leq C_{pu} < 2.000$ |
| Super | $2.000 \leq C_{pu}$ |

## 4. Test, Analysis Results and Discussions

This study proposed a plan to improve the environmental comfort and air quality in an office space based on data on the changes in temperature and humidity and the particulate matter concentrations, including $PM_{2.5}$, $PM_{10}$ and TSP, collected before the plans were implemented. Then, the improved strategies were proposed to mend the possible defects in this office and on-site experiments were subsequently conducted to compare data collected before and after the implementation of the improvement plan. The experimental method of this study was to establish a chronological timeline of the experimental data over an extended period. Then, statistical analysis methods were employed to compare the real improvement efficiency, and changes in indoor temperature, relative humidity, and the concentrations of $PM_{2.5}$ and $PM_{10}$ during the experimental period.

### 4.1. Variation of Temperature and Humidity Analysis

This study analyzed and compared the air quality before and after the implementation of the air quality improvement plan. Figure 5a shows that, during the first measurement period of 7 days, the temperature rose after sunrise when the air conditioning or ventilation system was deactivated, and it decreased gradually after sunset, showing that the temperature was 28–29 °C in the unused state. The humidity remained stable, at 60–65%. Between 6:00 and 18:00 (military time) on a normal working day, the temperature rose from 27 °C to 28 °C at sunrise, and after 8:00, when the air conditioning system was activated, it gradually decreased to 25 °C. After 18:00, the temperature rose gradually to 28 °C due to the transfer of the heat stored in the building shell to the indoor air. After 24:00, the temperature decreased to 27 °C. The humidity decreased significantly with the temperature, and while the air conditioning system was operating, the humidity remained at 45–50%. Therefore, improvements to the equipment were implemented as follows: (1) The mandatory ventilation system was increased, and (2) the damaged air exchange systems were partially repaired. Figure 5b shows that the temperature during working hours was mostly maintained at 26 °C. Compared with the temperature before the improvement, it was more stable. The relative humidity was maintained at around 55%, which was higher than before the improvement. This increase was caused by the higher rate of external air exchange, which reduced the difference between the internal and external humidity values.

To verify the improvements due to the implementation of the improvement plan, statistical analyses were conducted to evaluate the changes in the index values of the indoor temperature and relative humidity. The statistical values and comparison of the analysis results are listed in Table 2. The overall indicators of the improvement in temperature indicated that the measures were more effective in the staff seating area. The indicator values displayed a downward trend, showing specific improvements. However, the temperature was lower in the office equipment area than in the staff seating area. The main reason was that the temperature of the office machine area was kept low to maintain the normal operation of the photocopying machine. The statistical values of relative humidity showed that the overall relative humidity value fell within the comfort range, but the amount of improvement was less obvious. In particular, the daily numerical comparison revealed that the relative humidity of the photocopier area was high, mainly because the transaction

area and the operation process generated suspended particles, so the relative humidity was increased in the surrounding area to minimize the impact of the suspended particles.

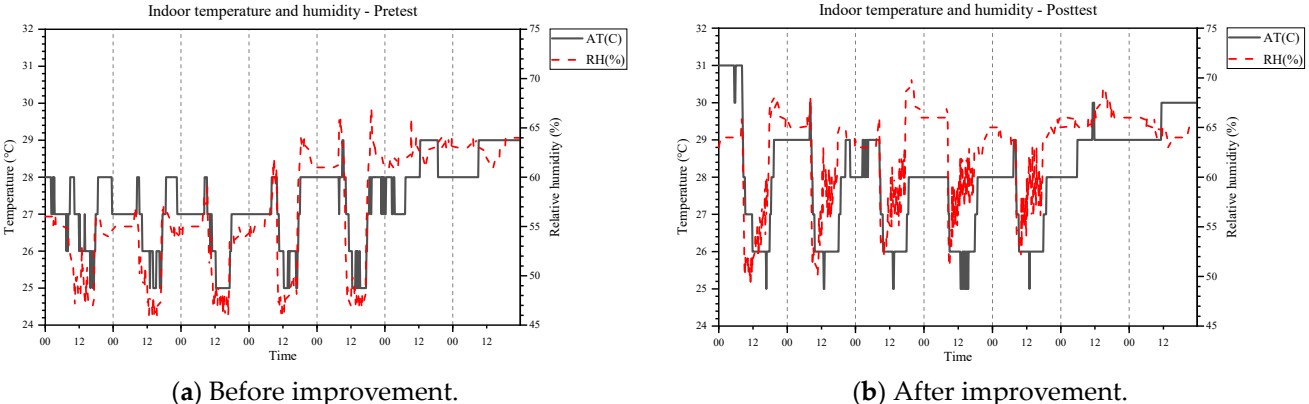

(**a**) Before improvement.　　　　　　　　(**b**) After improvement.

**Figure 5.** Analysis of the indoor temperature and humidity in the office space before and after the implementation of the improvement plan.

**Table 2.** Comparison of environmental performance indicators before and after the implementation of the improvement plan.

| Day | | | **Temp** | | | **RH** | | |
|---|---|---|---|---|---|---|---|---|
| | | | $\mu$ | $\sigma$ | $C_i$ | $\mu$ | $\sigma$ | $C_i$ |
| 1 | Before | A | 26.27 | 0.99 | 0.19 | 49.95 | 2.27 | 0.55 |
| | Change | B | 26.56 | 0.83 | 0.11 | 49.45 | 2.12 | 0.63 |
| | After | A | 25.05 | 1.47 | 0.77 | 57.93 | 3.47 | 0.37 |
| | change | B | 26.71 | 1.21 | 0.21 | 54.36 | 3.89 | 0.28 |
| 2 | Before | A | 25.16 | 1.23 | 0.62 | 50.85 | 2.58 | 0.42 |
| | Change | B | 26.13 | 0.86 | 0.18 | 49.09 | 3.02 | 0.78 |
| | After | A | 24.42 | 1.08 | 1.01 | 61.04 | 3.57 | 0.87 |
| | change | B | 26.36 | 1.02 | 0.18 | 56.65 | 4.19 | 0.36 |
| 3 | Before | A | 26.27 | 0.99 | 0.19 | 42.45 | 9.97 | 4.57 |
| | Change | B | 25.89 | 1.08 | 0.30 | 50.15 | 4.01 | 0.70 |
| | After | A | 24.33 | 1.09 | 1.08 | 61.78 | 2.92 | 0.97 |
| | change | B | 26.29 | 0.83 | 0.14 | 57.25 | 3.51 | 0.31 |
| 4 | Before | A | 25.16 | 1.23 | 0.62 | 36.75 | 3.63 | 6.16 |
| | Change | B | 26.27 | 1.06 | 0.21 | 50.76 | 4.99 | 0.76 |
| | After | A | 24.18 | 0.94 | 1.14 | 62.76 | 2.34 | 1.17 |
| | change | B | 26.02 | 0.71 | 0.17 | 58.36 | 3.57 | 0.43 |
| 5 | Before | A | 26.27 | 0.99 | 0.19 | 36.65 | 4.38 | 6.32 |
| | Change | B | 26.27 | 1.33 | 0.30 | 51.44 | 6.37 | 0.95 |
| | After | A | 24.40 | 0.83 | 0.95 | 62.00 | 1.91 | 0.94 |
| | change | B | 26.22 | 0.69 | 0.12 | 57.71 | 3.04 | 0.29 |
| 6 | Before | A | 26.27 | 0.99 | 0.19 | 41.29 | 0.46 | 3.34 |
| | Change | B | 28.51 | 0.50 | 0.43 | 62.45 | 0.83 | 1.00 |
| | After | A | 28.69 | 0.47 | 0.51 | 66.80 | 1.15 | 2.50 |
| | change | B | 29.07 | 0.26 | 0.71 | 66.58 | 1.45 | 2.42 |
| 7 | Before | A | 25.16 | 1.23 | 0.62 | 41.16 | 0.63 | 3.41 |
| | Change | B | 28.87 | 0.34 | 0.60 | 62.18 | 0.80 | 0.93 |
| | After | A | 29.29 | 0.74 | 0.93 | 64.38 | 1.16 | 1.59 |
| | change | B | 29.64 | 0.49 | 1.14 | 64.31 | 0.72 | 1.55 |

Comprehensive observation showed that this proposed strategy could provide a stable temperature and relative humidity, which improved the indoor comfort. This strategy would also reduce the effects of the day–night temperature difference on human emotions when the seasons change. Due to the compulsory ventilation and repairs to the external air

exchange, the temperature shock and changes in relative humidity were reduced to achieve more stable indoor comfort levels.

### 4.2. Particulate Matter Analysis

Particulate matter (PM) consists of coarse and fine particles of matter floating in the air, and the particles range in size depending on the source of the pollution. PM is composed of complex mixtures of organic and inorganic solids and liquids, and even water. These particles can penetrate deep into the alveoli and reach the membranous wall of the bronchi, where they interfere with the exchange of gas in the lungs due to the small particle size of fine PM ($PM_{2.5}$). Therefore, $PM_{2.5}$ is often listed as one of the key items of air quality measurement. Exposure to PM for a long time lead to cardiovascular disease, respiratory disease, and increased risk of lung cancer, particularly in sensitive populations, so it will cause great harm to both body and mind. In this study, the air quality meters set up at point A, at the electrical equipment, and at point B, at the first row of employee seats, monitored the amounts of suspended particulate matter, including $PM_{2.5}$, $PM_{10}$ and TSP, for comparison of the air quality before and after environmental improvement. The results of the statistical analysis are listed in Table 3. The variations in the suspended particulate matter are discussed below.

**Table 3.** Comparison of the indoor air quality indicators before and after the implementation of the improvement plan.

| Day | | | $PM_{2.5}$ | | | $PM_{10}$ | | | TSP | | |
|---|---|---|---|---|---|---|---|---|---|---|---|
| | | | $\mu$ | $\sigma$ | $C_{pu}$ | $\mu$ | $\sigma$ | $C_{pu}$ | $\mu$ | $\sigma$ | $C_{pu}$ |
| 1 | Before | A | 16.15 | 3.23 | −0.12 | 40.58 | 11.11 | 0.28 | 54.25 | 14.22 | 1.78 |
| | Change | B | 14.90 | 2.72 | 0.01 | 39.03 | 9.49 | 0.39 | 55.60 | 14.21 | 1.75 |
| | After | A | 13.35 | 2.26 | 0.24 | 21.95 | 5.25 | 1.78 | 26.82 | 6.38 | 5.39 |
| | change | B | 12.20 | 2.70 | 0.35 | 21.34 | 6.66 | 1.43 | 28.09 | 8.44 | 4.02 |
| 2 | Before | A | 23.39 | 57.64 | −0.05 | 102.83 | 446.75 | −0.04 | 139.26 | 591.23 | −0.01 |
| | Change | B | 16.32 | 18.53 | −0.02 | 48.06 | 84.19 | 0.01 | 67.34 | 103.06 | 0.20 |
| | After | A | 5.10 | 2.05 | 1.61 | 12.41 | 5.62 | 2.23 | 17.34 | 6.71 | 5.60 |
| | change | B | 4.56 | 1.97 | 1.77 | 13.13 | 6.74 | 1.82 | 21.51 | 9.75 | 3.71 |
| 3 | Before | A | 13.33 | 7.39 | 0.08 | 29.29 | 17.19 | 0.40 | 42.57 | 22.77 | 1.28 |
| | Change | B | 13.07 | 7.42 | 0.09 | 29.83 | 16.94 | 0.40 | 45.24 | 20.69 | 1.37 |
| | After | A | 4.02 | 0.68 | 5.39 | 9.68 | 2.06 | 6.53 | 14.49 | 3.33 | 11.55 |
| | change | B | 3.57 | 0.61 | 6.28 | 10.40 | 4.90 | 2.69 | 18.45 | 9.93 | 3.75 |
| 4 | Before | A | 15.97 | 6.13 | −0.05 | 30.75 | 14.63 | 0.44 | 41.48 | 17.11 | 1.72 |
| | Change | B | 16.48 | 8.26 | −0.06 | 37.16 | 26.57 | 0.16 | 54.65 | 35.68 | 0.70 |
| | After | A | 4.15 | 1.75 | 2.07 | 10.57 | 9.21 | 1.43 | 15.04 | 12.52 | 3.06 |
| | change | B | 3.64 | 1.49 | 2.54 | 11.62 | 11.70 | 1.09 | 19.29 | 16.71 | 2.21 |
| 5 | Before | A | 12.51 | 9.82 | 0.08 | 27.96 | 18.42 | 0.40 | 40.05 | 19.99 | 1.50 |
| | Change | B | 12.01 | 9.23 | 0.11 | 29.11 | 18.09 | 0.38 | 44.89 | 21.07 | 1.35 |
| | After | A | 3.27 | 0.83 | 4.73 | 8.02 | 1.85 | 7.57 | 12.47 | 3.68 | 10.65 |
| | change | B | 2.91 | 0.78 | 5.14 | 8.66 | 3.44 | 4.01 | 16.21 | 9.45 | 4.02 |
| 6 | Before | A | 18.37 | 3.90 | −0.29 | 22.48 | 5.59 | 1.64 | 22.73 | 5.66 | 6.32 |
| | Change | B | 17.96 | 4.36 | −0.23 | 22.84 | 7.48 | 1.21 | 23.43 | 8.91 | 3.99 |
| | After | A | 8.53 | 4.11 | 0.52 | 11.21 | 6.38 | 2.03 | 11.90 | 7.06 | 5.58 |
| | change | B | 8.13 | 3.94 | 0.58 | 10.60 | 6.09 | 2.16 | 11.30 | 6.87 | 5.76 |
| 7 | Before | A | 14.00 | 2.06 | 0.16 | 16.54 | 3.39 | 3.29 | 16.60 | 3.40 | 11.12 |
| | Change | B | 13.22 | 1.78 | 0.33 | 15.72 | 2.89 | 3.96 | 15.79 | 2.85 | 13.34 |
| | After | A | 6.98 | 1.23 | 2.17 | 9.89 | 2.77 | 4.83 | 10.37 | 3.10 | 12.87 |
| | change | B | 6.76 | 1.13 | 2.42 | 9.34 | 2.30 | 5.88 | 9.86 | 2.69 | 14.86 |

### 4.2.1. Comparison of $PM_{2.5}$ between before and after Improvement

Figure 6a shows that the $PM_{2.5}$ concentration during the monitored time was highest from approximately 10:00 to 12:00. It was observed that the electrical equipment was frequently used in this time period. Then, the $PM_{2.5}$ concentration fell significantly because

the machines were deactivated during the lunch break at noon before gradually increasing when these machines were reactivated at 2 p.m. The maximum $PM_{2.5}$ concentration values measured during the monitoring period were 30–60 $\mu g/m^3$, and the lowest concentration was around 5 $\mu g/m^3$. If the $PM_{2.5}$ concentration is greater than 25 $\mu g/m^3$ for more than 3 days per year, the air quality is listed as "Bad", according to the World Health Organization (WHO) standards. To qualify as "Good", the $PM_{2.5}$ concentration must be less than 12 $\mu g/m^3$, according to the U.S. Environmental Protection Agency [32], or 15 $\mu g/m^3$, according to the Environmental Protection Administration of Taiwan [33]. Figure 6a shows that, before the implementation of the air quality improvement plan, the $PM_{2.5}$ concentration in this space exceeded the standards set by the Environmental Protection Agency (EPA) of Taiwan during a portion of the time period, did not meet the "good" standard of the U.S. Environmental Protection Agency, and even met the WHO's "bad" standard when the electrical equipment was in use. The test data measured at point A were almost identical to those at point B (worker's seats), showing that the concentration of suspended particulate matter produced by the electrical equipment directly affected the office seating area.

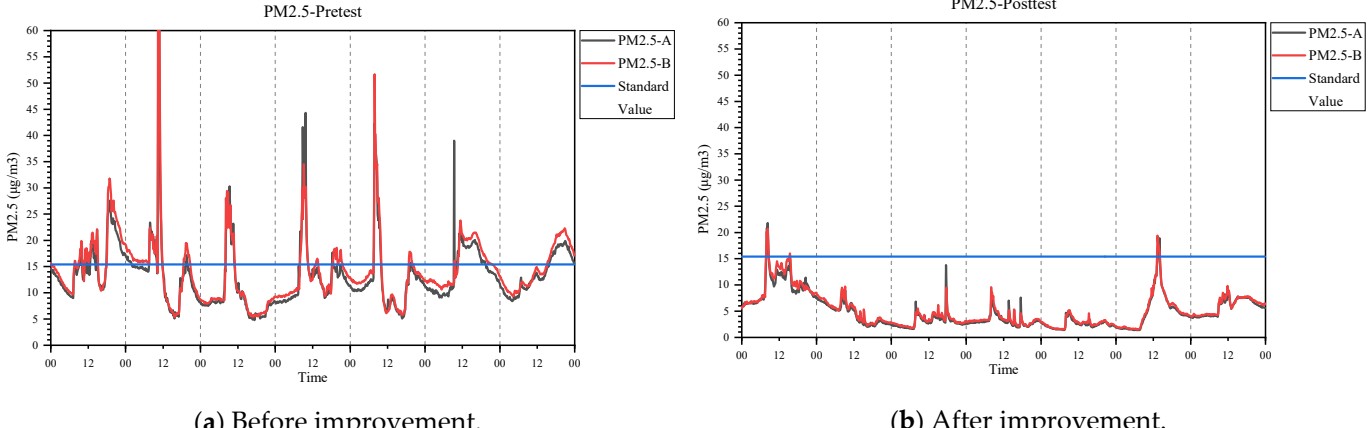

(**a**) Before improvement.          (**b**) After improvement.

**Figure 6.** Comparison of indoor $PM_{2.5}$ concentrations before and after implementation of the improvement plan.

Based on the monitoring and analysis results, an improvement plan was proposed as follows: (1) clean the air conditioning filter; (2) create openings in the ceiling and install air filters; and (3) add ventilation systems. After these changes, the $PM_{2.5}$ concentration was measured for comparison with that before the implementation of the improvement plan and also to assess whether the air quality in the office space was improved. The results of the test conducted after the changes are shown in Figure 6b As can be seen in that figure, the overall $PM_{2.5}$ concentration was significantly reduced. The highest $PM_{2.5}$ concentration value during the monitoring period was 13–23 $\mu g/m^3$, and the lowest $PM_{2.5}$ concentration was reduced to 3 $\mu g/m^3$.

Variations in the $PM_{2.5}$ concentration, an indicator of poor indoor air quality, are listed in Table 3. The data show that the $PM_{2.5}$ concentration was reduced by the strategies implemented. Notably, on four days, the air quality achieved "Super" ratings; on one day, "Excellent"; and on only two days, both of which were Saturdays, "Inadequate". Overall, the $PM_{2.5}$ concentration was reduced to less than 15 $\mu g/m^3$, the specification set by Taiwan's EPA.

Test results have shown that cleaning an air filter increases the rate of air circulation, and also that ceilings with more openings and installed air filters increase the filtering efficiency of the overall space when coupled with a compulsory ventilation system. Such a design prevents the continuous accumulation of particles in indoor blind spots. The test results showed that the indoor $PM_{2.5}$ concentration could be decreased effectively and rapidly through these implemented measures.

### 4.2.2. The Comparison of $PM_{10}$ Concentration before and after Improvement

$PM_{10}$ particles are large aerosols. They easily affect people with sensitive health conditions, and long-term exposure will contribute to cardiopulmonary and other related diseases. In the original office environment, the air conditioning equipment was not regularly cleaned, so the airflow in the space was impeded. As a result, the $PM_{10}$ particles could not easily be eliminated by filtration and air exchange. In addition, the office space had numerous blind spots, so $PM_{10}$ particles could remain for years in the office partitions, bookcases, or even carpets. Figure 7a shows that the trend of variation in the $PM_{10}$ concentration was the same as that of $PM_{2.5}$. Roughly, the levels exceeded the standard value of $PM_{10}$ from 10:00 to 12:00. The highest $PM_{10}$ concentrations were 80–180 $\mu g/m^3$, well exceeding the $PM_{10}$ standard value of Taiwan's EPA, 50 $\mu g/m^3$. The difference from Figure 6 is that the jerk values in the range of 20–60 $\mu g/m^3$ were also monitored during the working hours outside the 10:00–12:00 time slot.

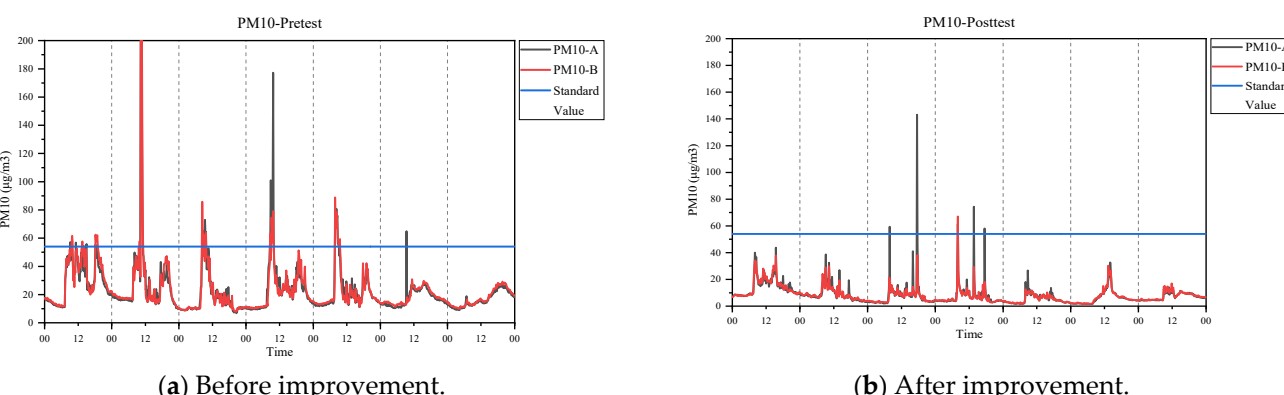

(**a**) Before improvement.                                                  (**b**) After improvement.

**Figure 7.** Comparison of indoor $PM_{10}$ concentrations before and after the implementation of the improvement plan.

Investigation and analysis of the indoor space and environmental quantity of this office revealed that the heavy carpets in the office had not been fully cleaned in several years. The condition of the carpets increased the accumulation of suspended particles in the air, and people walking on the carpet would stir up dust and thus increase the $PM_{10}$ concentration.

Therefore, based on the monitoring and analysis results, an improvement plan was proposed as follows: (1) remove the office partitions to install partitions with better ventilation so as to increase the air flow and eliminate blind spots; (2) replace the heavy carpet to reduce dust accumulation; and (3) install forced ventilation to directly prevent the impacts of pollution and accelerate the air exchange. After the implementation of this plan, the $PM_{10}$ concentration was monitored for comparison with the data collected before the changes. Test results from after the changes are shown in Figure 7b. Figure 7b shows that the $PM_{10}$ concentration was significantly lower during the monitoring period. The highest concentrations were 60–140 $\mu g/m^3$. In addition, the jerk value during office hours was gradually reduced, and the minimum $PM_{10}$ concentration was about 5 $\mu g/m^3$. The $PM_{10}$ concentrations listed in Table 3 show that the $PM_{10}$ concentrations were lowered by the improvement plan. Notably, the $PM_{10}$ concentrations all exceeded the "excellent" quality conditions, even on weekends, when the air conditioning system was deactivated. Overall, the concentration was reduced to less than 15 $\mu g/m^3$, the specification set by Taiwan's EPA.

These test results showed that the blind spots and accumulated dust were significantly reduced through carpet removal and the replacement of the partitions to improve ventilation.

### 4.2.3. Monitoring and Analysis of Total Suspended Particulates (TSP)

The test results of total suspended particulates (TSP) were shown in Figure 8a. This figure revealed the overall suspension particles variation during test period. Figure 8a displayed that the TSP increased rapidly from 7 a.m. of office hours due to human activity and

electrical equipment use and decreased gradually by 6 p.m. when they got off. Otherwise, TSP concentration maintained at the lowest and stable condition at the weekend because a small number of people work overtime. The highest concentrations in the experimental period were the same as the those of $PM_{2.5}$ and $PM_{10}$ trends, occurring at 10–12 a.m. There were heavily printed, so the photocopier equipment was used uninterrupted for a long time, so the concentration rose rapidly and highly, with the highest measured concentration between 100–280 $\mu g/m^3$. On the other hand, the TSP variations revealed that TSP concentration beat between 20–60 $\mu g/m^3$ due to electrical equipment used. The monitoring curves of point A and B were similar before office improved, indicating that the TSP concentration of the electrical equipment area would directly affect the seating area of the working colleagues. After the implementation of the improvement scheme, Figure 8b displayed that TSP concentration curve of point A was higher than that of point B. Despite the reduction in TSP concentration at point A after improvement, the electrical equipment still produced a partial concentration of suspended particulates. These TSP were excluded quickly through air flow by the updated forced ventilation or ventilation equipment. The maximum concentration measured at point A after improvement was roughly reduced to between 80–110 $\mu g/m^3$, with only the maximum of 190 $\mu g/m^3$. The rest of test results were significantly lower than those before the improvement. It was worth noting that even if the TSP concentration of point A was as high as 190 $\mu g/m^3$, the TSP concentration of point B was only raised to about 50 $\mu g/m^3$, which was below the standard value of 130 $\mu g/m^3$. The TSP variation of indicator of indoor air quality, listed in Table 3, showed that air quality condition of TSP attained "Super" quality conditions, including weekends, in which the air conditioner was closed. The improved benefits of these proposed strategies have been verified.

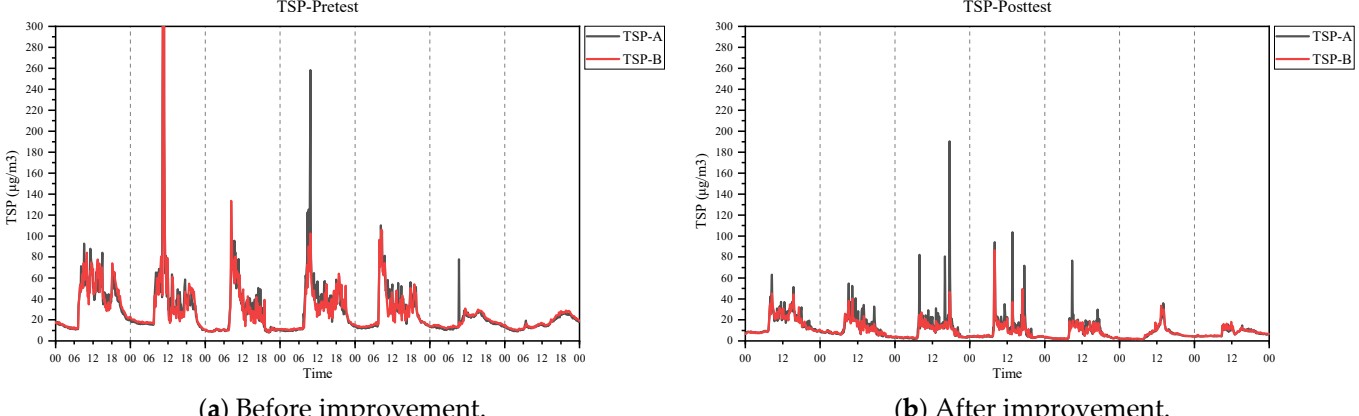

(**a**) Before improvement.                    (**b**) After improvement.

**Figure 8.** Comparison of indoor TSPs before and after the implementation of the improvement plan.

## 5. Comprehensive Analysis and Discussions

The previous monitoring data were applied to identify the influence factors of the indoor air quality, temperature and relative humidity of this office space in this study. Then, the main research purpose of this study was to find the most economical way to improve the indoor environment and maintain fine air indoor air quality. The statistical analysis of the detected data before and after improvement of this office space was discussed as follows:

### 5.1. The Improved Efficiency of Indoor Temperature and Relative Humidity

The analysis results of environmental performance indicator in Table 2 showed the indoor temperature and relative humidity variance of office space before and after changes, it was found that:

1.      Seating area:

Indoor temperature at working hours: The average indicator value before the improvement ranged from 0.3 to 0.11. Then, the indicator value was between 0.12 and 0.21 after the

improvement, the average indicator value decreased from 0.22 to 0.16. The indicator value of relative humidity was between 0.63 and 0.95 before the improvement and decreased to 0.28 and 0.43 after changes. Especially, the average indicator of relative humidity dropped from 0.76 to 0.33.

The statistical analysis characteristics of the environmental performance indicator was the smaller-the-best. These values revealed that these improved strategies specifically improve the indoor temperature and relative humidity of the seating area.

2.    Photocopier area:

Temperature change during business hours in this area: In order to maintain the operation of the photocopy machine, the temperature must be lower than the seating area to avoid the increase in the ambient temperature due to the operation of the transaction machine. The indicator value in this area ranged from 0.99 to 0.19 before changes and increased to between 1.14 and 0.77 after improved. The average indicator value raised from 0 36 to 0.99, which should be maintained at a lower temperature than the seating area. Comparing the temperature changes, it could be found that the average temperature before the improvement in this area was between 26.27 °C and 25.16 °C, and the average temperature after the improvement was between 25.16 °C and 24.18 °C. The average temperature was reduced from 25.83 °C to 24.48 °C. The indicator value of the relative humidity in this area was between 6.32 to 0.42 before change, and these indicator values reduced to between 1.17 and 0.37. The average indicator value decreased from 3.60 to 0.86 after changes. From the statistical analysis characteristics of this performance indicator, the analysis results displayed that these improved strategies could effectively reduce the ambient temperature of the photocopy area and maintain appropriately relative humidity around this area.

The difference in the indicator of temperature and relative humidity cannot be compared with those before and after changes because the air-conditioning equipment cannot be turned on by the government regulations.

### 5.2. Improving Efficiency for Indoor Air Quality

Regarding the variation of $PM_{2.5}$, $PM_{10}$ and TSP concentrations, the statistical analysis results are shown in Table 3. These indicators of indoor air quality variance in this office space before and after changes, it was found that:

1.    Seating area:
(1)    Indicator of indoor air quality for $PM_{2.5}$ concentrations:

Office Hours: The average indicator value of weekdays before the improvement was between −0.05 and 0.11; and raised to between 0.35 and 6.28 after changes. The average indicator values of office hours increased from 0.02 to 3.22. The quality condition for the improving efficiency of these proposed strategies reached to the "Super" quality condition.

Holidays and weekends: Even if only a few people worked overtime on holidays or weekends, the average indicator value from 0.05 increased to 1.5, attained to the "Satisfactory" quality condition.

(2)    Indicator of indoor air quality for $PM_{10}$ concentrations:

Office Hours: The average indicator values of weekdays before improvement ranged from 0.01 to 0.39 and improved to between 1.09 and 4.01. The average indicator of office hours rose from 0.27 to 2.21. The quality condition of the improving efficiency of these proposed strategies for PM 10 concentrations reached to the "Super" quality condition.

Holidays and weekends: Even if only a few people work overtime on holidays and weekends, the average indicator value was from 2.45 increased to 4.02.

(3)    Indicator of indoor air quality for TSP concentrations:

Office Hours: The average indicator values before changes ranged from 0.02 to 1.75 and increased between 2.21 and 4.02 after improvement. The average indicator of office

hours raised from 0.91 to 3.54. The quality condition of the improving efficiency of these proposed strategies for TSP concentrations reached to the "Super" quality condition.

Holidays and weekends: Even if only a few people work overtime on holidays and weekends, these average indicator values reached to "Super" quality condition.

Actually, the quality condition of the improving efficiency for $PM_{10}$ concentrations and TSP concentrations on holidays and weekends all attained the "Super" quality condition. The change in the average indicator value cannot show its specific improvement benefits on holidays and weekend.

2.  Photocopier area:
(1)  Indicator of indoor air quality for $PM_{2.5}$ concentrations:

Office Hours: The average indicator value before the improvement was between −0.12 and 0.08, and the average indicator value after changes was between 0.24 and 5.39. The average indicator of the whole office hours increased from −0.01 to 2.82. The quality condition of the improving efficiency of these proposed strategies for $PM_{2.5}$ concentrations reached to the "Super" quality condition.

Holidays and weekends: there were only a few people that work overtime on holidays and weekends, so average indicator value of holidays and weekends was raised from −0.07 to 1.34, attaining the "Satisfactory" quality condition.

(2)  Indicator of indoor air quality for $PM_{10}$ concentrations:

Office Hours: The average indicator value before changes was between −0.04 and 0.44, and the average indicator value increased to between 1.43 and 7.57. The average indicator value of the whole office hours rose from 0.15 to 3.91. The quality condition of the improving efficiency of these proposed strategies for $PM_{10}$ concentrations reached the "Super" quality condition. Holidays and weekends: Even if only a few people work overtime on holidays, the average indicator value of weekends and holidays was between 2.47 and 3.43.

(3)  Indicator of indoor air quality for TSP concentrations:

Office hours: The average indicator value before changes was between −0.01 and 1.78, and the average indicator value before changes increased to between 3.06 and 11.55. The average indicator of the whole office hours rose from 1.25 to 7.25. The quality condition of the improving efficiency of these proposed strategies for TSP concentrations reached to the "Super" quality condition.

Holidays and weekends: Even if only a few people work overtime on holidays, the average indicator value of holiday and weekend changes also reached the "Super" quality condition

The improved efficiency of $PM_{10}$ concentrations and TSP concentrations before and after changes at the transaction area was consistent with the seating area. The specific improvement benefits of $PM_{10}$ concentrations and TSP concentrations could not be revealed on holidays and weekends.

### 5.3. Fraction Defective Analysis

In this study, the total number of sampling was 385 for one week. The $PM_{2.5}$ and PM 10 concentrations of Taiwan canonical value was 15 μg/ m³ and 50 μg/ m³, respectively. The defect rate was calculated by deducting those exceeding the standard value from the total sample size, and the relevant values are summarized as shown in Table 4. The analysis results of Table 4 showed that the Fraction Defective of $PM_{2.5}$ and $PM_{10}$ reduced from 35.58% to 2.08% and from 8.57% to 0.52%, respectively. After changes in seating area, the proportion of Fraction Defective of $PM_{2.5}$ concentrations has been reduced from 40.78% to 2.60% and $PM_{10}$ from 8.31% to 0.26%, respectively. These results revealed that these improved strategies effectively solve the situation of indoor $PM_{2.5}$ and $PM_{10}$ concentrations defects. The distribution curves with the $PM_{2.5}$ and $PM_{10}$ concentration as the X axis and the

number of monitored sampling as Y axis were shown in Figures 9 and 10. Figures 9 and 10 displayed that the

**Table 4.** Comparison of unqualified ratios before and after the implementation of the improvement plan.

| | Test Point | The Total Number of Samples (A) | Standardize Standard Values (B) | Before Improvement More Than (B). Number of Samples (C) | Improved More Than (B). The Number of Samples (D). | Before Change (Unqualified Ratio) C/A×100 | After Change (Unqualified Ratio) D/A×100 |
|---|---|---|---|---|---|---|---|
| PM$_{2.5}$ | A | 385 | 15 | 137 | 8 | 35.58% | 2.08% |
| | B | 385 | 15 | 157 | 10 | 40.78% | 2.60% |
| PM$_{10}$ | A | 385 | 50 | 33 | 2 | 8.57% | 0.52% |
| | B | 385 | 50 | 32 | 1 | 8.31% | 0.26% |

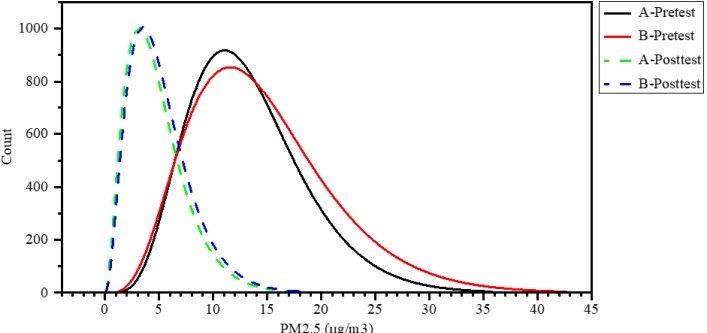

**Figure 9.** Distribution curve of PM$_{2.5}$ concentrations before and after the implementation of the improvement plan.

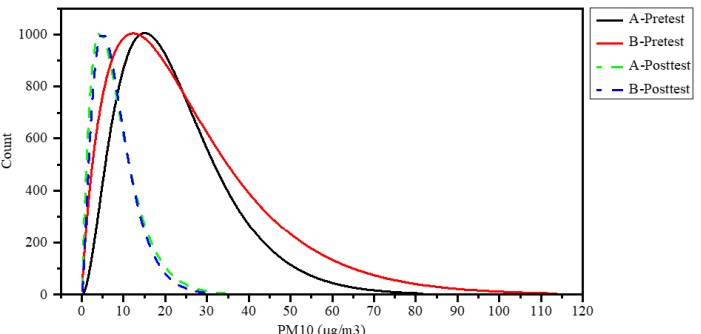

**Figure 10.** Distribution curve of PM$_{10}$ concentrations before and after the implementation of the improvement plan.

PM$_{2.5}$ and PM$_{10}$ concentration change interval after change in the photocopy area and the seating area has been narrowed, and the average maximum of PM$_{2.5}$ and PM$_{10}$ concentration has been reached below the standard value. The most commonly measured PM$_{2.5}$ concentration has been reduced to 3 µg/ m$^3$. The most commonly measured concentration of PM$_{10}$ has been reduced to 5 µg/ m$^3$. The photocopy area showed the same trend as the seating area. From Figures 9 and 10′s waveform trend distribution, it was seen that the PM$_{2.5}$ and PM$_{10}$ variation ranges in the seating area were mainly photocopiers for toner laser printers, and the fine suspended particles produced by photocopying were susceptible to wind speed and direction. Susceptibility to airflow results in a wide distribution of these curves. However, after the implementation of this scheme, in addition to effectively reducing the distribution of suspended particles. Fine suspended particles were quickly removed by air exchange and forced ventilation.

## 6. Conclusions

This study focused on the commonly used office space in Taiwan, has a seating area, a document storage area, and a photocopying machine area. To avoid the outdoor pollution to affect the indoor air quality, the full air conditioning system was applied to this kind office space. After the initial experimental analysis, some defects in this kind office space have been found and then to propose some strategies to improve the indoor environment and indoor air quality. The improved efficiency of these strategies would be calculated by the statistical method based on the experimental data; the following conclusions were obtained as follows:

1.  By improving the forced ventilation system and external air exchange system, the indoor temperature and relative humidity maintained around 26 °C and 55%, respectively, during office hours.
2.  The improved efficiency of temperature and relative humidity increased 27.27% and 56.56% for the seating area and 66.67% and 76.11% for the photocopier area, respectively.
3.  The improved benefits of air quality for the seating area using the indicator of indoor air quality for $PM_{2.5}$, PM 10 and TSP during working period reached to the "Satisfactory" quality condition, the "Super" quality condition and the "Super" quality condition, respectively.
4.  The improved benefits of air quality for the photocopying area using the indicator of indoor air quality for $PM_{2.5}$, $PM_{10}$ and TSP during working period all reached to the "Super" quality condition.
5.  The analysis results showed that the most commonly measured $PM_{2.5}$, $PM_{10}$ concentration in photocopier area has been decreased from 11 µg/ $m^3$ to 3 µg/ $m^3$ and from 20 µg/ $m^3$ to 5 µg/ $m^3$, respectively through the total number of sampling.

The test and analysis results of this study display that the implementation of these improved strategies effectively improve the indoor environment and reduce the generation of suspended particles. These strategies were used as a reference for the other relevant offices to modify the indoor environment and air quality of office spaces.

**Author Contributions:** Conceptualization, T.-Y.C., W.-P.S. and C.-H.L.; methodology, W.-P.S. and C.-H.L.; software, T.-Y.C., W.-P.S.; formal analysis, W.-P.S. and T.-Y.C.; data curation, W.-P.S. and C.-H.L.; writing—original draft preparation, T.-Y.C., W.-P.S. and C.-H.L.; writing—review and editing, T.-Y.C., W.-P.S. and C.-H.L.; visualization, T.-Y.C., W.-P.S. and C.-H.L.; project administration, W.-P.S.; funding acquisition, W.-P.S. All authors have read and agreed to the published version of the manuscript.

**Funding:** This research was funded by Ministry of Science and Technology, Taiwan, grant number No. MOST-110-2410-H-167-002-MY2.

**Institutional Review Board Statement:** Exclude this statement. This study did not require ethical approval.

**Informed Consent Statement:** Not applicable.

**Data Availability Statement:** All data are available within the article and also from the corresponding author upon request.

**Conflicts of Interest:** The authors declare that there is no conflict of interest regarding the publication of this paper.

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
