# Peer review of "Strategy for Improving the Indoor Environment of Office Spaces in Subtropical Cities"

_buildings, doi:10.3390/buildings12040412_

Round 1

Reviewer 1 Report

I would like to thank the authors for the major improvements, the manuscript readability and quality have improved. Although the manuscript has improved yet the manuscript lacks scientific soundness. Fundamental flaws in the methodology of the study can be seen that has not been addressed by authors. The results and discussion need to be improved significantly. Yet, the number of references is not sufficient, and the introduction needs to be improved. I believe the manuscript is not publishable in this current format. Here are my additional comments to improve the manuscript.

  • Although the abstract has been improved it lacks scientific soundness. In the last three sentences, instead of using vague terms such as “significantly improved”, statistical data should be given to defining the improvement.
  • The last sentence of the introduction should be rewritten. What do ya mean by “otherwise”?
  • The study gap and contribution of this study to knowledge is still missing.
  • “environmental improvement performance indicator [18, 19] was quoted in this study to evaluate the improvement benefits of indoor temperature and humidity before and after the improvement.” This sentence has been reported frequently in the manuscript. This is redundant and reduces the quality of the manuscript. Also, there is no need to use bold words in the manuscript.
  • Line 90, “24.2~29.5˚C” selected suitable range is based on a study of classrooms, which is different from the office building. Also, an optimal temperature can be defined according to the type of ventilation and cooling system or natural ventilation. In any condition, higher temperatures are not advisable as mentioned in various studies in the literature. I find it difficult to realize any reasonable justification for this suggested temperature range.
  • Line 168, how the interview was conducted? How many interviews? What was the question? On what basis the results are concluded? What was the Sick building syndrome symptoms of the interview? For how long these symptoms were monitored?
  • As commented previously, what is the justification of the locations of the sensors adjacent to the printer? The equipment and loggers should be close to the sitting area of a person to give a reliable value of a breathable area.
  • The hours to report the air temperature and relative humidity should be related to working hours or the tine HVAC system is on or off. At this point, the compression of these two variables does not give any significant findings.
  • The symbols in Table 2 should be defined. In table 2, on some days there is no significant difference between pre and after tests but some days shows higher differences. This inconsistent data makes the findings to be unreliable. Why is this happening?
  • What kind of statistical test is used to compare the data?
  • The study lacks strong discussion to compare the findings of this study with other studies. The authors should read similar articles and compare the findings with the current study.
  • The contribution of the study should be highlighted in the discussion which is missing.
  • Part of the conclusion is not supported by the data regarding the temperature and humidity to be improved “greatly”. There is no justification for this claim. Lower or higher temperature is not an indicator of an improved temperature but the subject's report can be in the index if there is a statistical comparison between occupants' feedback to different temperatures.

Author Response

Dear Sir/ Madam:

The answer for point to point of comments of Manuscript ID buildings-1625161 The Strategy for Improving Indoor Environment of Office Space in subtropical City has been modified following the reviewers’ comment. is as follows. Please check it again and see if it is acceptable for publication on Buildings.

Comments and Suggestions for Authors

I would like to thank the authors for the major improvements, the manuscript readability and quality have improved. Although the manuscript has improved yet the manuscript lacks scientific soundness. Fundamental flaws in the methodology of the study can be seen that has not been addressed by authors. The results and discussion need to be improved significantly. Yet, the number of references is not sufficient, and the introduction needs to be improved. I believe the manuscript is not publishable in this current format. Here are my additional comments to improve the manuscript.

  • Although the abstract has been improved it lacks scientific soundness. In the last three sentences, instead of using vague terms such as “significantly improved”, statistical data should be given to defining the improvement.

ANS: Thank you for your kind suggestion. We did rewrite the abstract. Please refer to this revised paper.

  • The last sentence of the introduction should be rewritten. What do ya mean by “otherwise”?

ANS: Thank you for your kind suggestion. We did rewrite the introduction section. Please refer to this revised paper.

  • The study gap and contribution of this study to knowledge is still missing.

ANS: Thank you for your kind suggestion. We did rewrite and organize the structure of this paper. Please refer to this revised paper.

  • “environmental improvement performance indicator [18, 19] was quoted in this study to evaluate the improvement benefits of indoor temperature and humidity before and after the improvement.” This sentence has been reported frequently in the manuscript. This is redundant and reduces the quality of the manuscript. Also, there is no need to use bold words in the manuscript. 

ANS: Thank you for your kind suggestion. We did modify them. Please refer to this revised paper.

  • Line 90, “24.2~29.5˚C” selected suitable range is based on a study of classrooms, which is different from the office building. Also, an optimal temperature can be defined according to the type of ventilation and cooling system or natural ventilation. In any condition, higher temperatures are not advisable as mentioned in various studies in the literature. I find it difficult to realize any reasonable justification for this suggested temperature range.

ANS: According to the provisions of Taiwan's "Comprehensive Energy-saving and Carbon Reduction Measures for Government Agencies and Schools”:

Air conditioning must be used in accordance with:

(1) Adopt responsible partition management, control the temperature of the office, conference room and classroom, set the appropriate temperature of the air conditioner (26 ~ 28 ° C), and use it with electric fans as needed.

(2) Holidays and weekends or a small number of people working overtime do not use the central air conditioning air conditioning.

Therefore, the standard for the control of temperature and relative humidity in the offices of public servants of Taiwan government is based on this study.

Line 168, how the interview was conducted? How many interviews? What was the question? On what basis the results are concluded? What was the Sick building syndrome symptoms of the interview? For how long these symptoms were monitored?

ANS: The study was conducted in the form of a questionnaire survey, with 38 people in the office, and a total of 38 questionnaires were issued with a 100% recovery rate. The questionnaire included the feeling of indoor temperature, humidity and indoor air quality, as well as the actual feelings of the office space. The physical reactions were: Mucosal symptoms: dry and itchy eyes and dry itchy throat; Skin symptoms: dry, itchy and reddened skin; Neurotoxic symptoms: headache, lethargy, and difficulty concentrating; Nonspecific symptoms: nasal congestion, runny nose, and wheezing.

The questionnaire results showed that 86% employee had the above-mentioned symptoms. Only 12% of those who had the above symptoms after improvement.

  • As commented previously, what is the justification of the locations of the sensors adjacent to the printer? The equipment and loggers should be close to the sitting area of a person to give a reliable value of a breathable area.

ANS: Thank you for your kind suggestion. We added the floor plan to mark the sensor locations. Please refer to this revised paper.

  • The hours to report the air temperature and relative humidity should be related to working hours or the tine HVAC system is on or off. At this point, the compression of these two variables does not give any significant findings.

ANS: Thank you for your kind assistance. We did rewrite the analysis section of this revised paper, focused on the working hours. Please refer to this revised paper.

  • The symbols in Table 2 should be defined. In table 2, on some days there is no significant difference between pre and after tests but some days shows higher differences. This inconsistent data makes the findings to be unreliable. Why is this happening?

ANS: Thank you for your kind suggestion. It is very useful for us to modify the presentation of this paper. We did rewrite the Comprehensive analysis and discussions and conclusions. Please refer to this revised paper.

  • What kind of statistical test is used to compare the data?

ANS: Thank you for your kind suggestion. The test results before and after changes were used the statistical method to compare the improving efficiency of the most economical way. Please refer to the analysis results of this revised paper.

  • The study lacks strong discussion to compare the findings of this study with other studies. The authors should read similar articles and compare the findings with the current study.

ANS: Thank you for your kind suggestion. It is very useful for us to modify the presentation of this paper. We tried to compare with the other research findings. Most of those results would took a lot of money to modify the indoor temperature, relative humidity and indoor air quality such as: added more air-conditioners, external sunshade systems and water spray systems and so on. Therefore, we did rewrite the Comprehensive analysis and discussions and conclusions. Please refer to this revised paper.

  • The contribution of the study should be highlighted in the discussion which is missing.

ANS: Thank you for your kind suggestion. It is very useful for us to modify the presentation of this paper. We did rewrite the Comprehensive analysis and discussions and conclusions. Please refer to this revised paper.

  • Part of the conclusion is not supported by the data regarding the temperature and humidity to be improved “greatly”. There is no justification for this claim. Lower or higher temperature is not an indicator of an improved temperature but the subject's report can be in the index if there is a statistical comparison between occupants' feedback to different temperatures.

ANS: Thank you for your kind suggestion. We did rewrite the conclusion section to highlight the finding of this paper. Please refer to this revised paper.

These comments are very useful for us to modify this paper. We are deeply appreciated for your kind suggestion and assistance. Please refer to this revised paper and review this paper. If there is any question, please feel free to contact me directly. I am looking forward to hearing from you soon. 

Best regards,

Wen-Pei Sung

Dear Sir/ Madam:

The answer for point to point of comments of Manuscript ID buildings-1625161 The Strategy for Improving Indoor Environment of Office Space in subtropical City has been modified following the reviewers’ comment. is as follows. Please check it again and see if it is acceptable for publication on Buildings.

Comments and Suggestions for Authors

I would like to thank the authors for the major improvements, the manuscript readability and quality have improved. Although the manuscript has improved yet the manuscript lacks scientific soundness. Fundamental flaws in the methodology of the study can be seen that has not been addressed by authors. The results and discussion need to be improved significantly. Yet, the number of references is not sufficient, and the introduction needs to be improved. I believe the manuscript is not publishable in this current format. Here are my additional comments to improve the manuscript.

  • Although the abstract has been improved it lacks scientific soundness. In the last three sentences, instead of using vague terms such as “significantly improved”, statistical data should be given to defining the improvement.

ANS: Thank you for your kind suggestion. We did rewrite the abstract. Please refer to this revised paper.

  • The last sentence of the introduction should be rewritten. What do ya mean by “otherwise”?

ANS: Thank you for your kind suggestion. We did rewrite the introduction section. Please refer to this revised paper.

  • The study gap and contribution of this study to knowledge is still missing.

ANS: Thank you for your kind suggestion. We did rewrite and organize the structure of this paper. Please refer to this revised paper.

  • “environmental improvement performance indicator [18, 19] was quoted in this study to evaluate the improvement benefits of indoor temperature and humidity before and after the improvement.” This sentence has been reported frequently in the manuscript. This is redundant and reduces the quality of the manuscript. Also, there is no need to use bold words in the manuscript. 

ANS: Thank you for your kind suggestion. We did modify them. Please refer to this revised paper.

  • Line 90, “24.2~29.5˚C” selected suitable range is based on a study of classrooms, which is different from the office building. Also, an optimal temperature can be defined according to the type of ventilation and cooling system or natural ventilation. In any condition, higher temperatures are not advisable as mentioned in various studies in the literature. I find it difficult to realize any reasonable justification for this suggested temperature range.

ANS: According to the provisions of Taiwan's "Comprehensive Energy-saving and Carbon Reduction Measures for Government Agencies and Schools”:

Air conditioning must be used in accordance with:

(1) Adopt responsible partition management, control the temperature of the office, conference room and classroom, set the appropriate temperature of the air conditioner (26 ~ 28 ° C), and use it with electric fans as needed.

(2) Holidays and weekends or a small number of people working overtime do not use the central air conditioning air conditioning.

Therefore, the standard for the control of temperature and relative humidity in the offices of public servants of Taiwan government is based on this study.

Line 168, how the interview was conducted? How many interviews? What was the question? On what basis the results are concluded? What was the Sick building syndrome symptoms of the interview? For how long these symptoms were monitored?

ANS: The study was conducted in the form of a questionnaire survey, with 38 people in the office, and a total of 38 questionnaires were issued with a 100% recovery rate. The questionnaire included the feeling of indoor temperature, humidity and indoor air quality, as well as the actual feelings of the office space. The physical reactions were: Mucosal symptoms: dry and itchy eyes and dry itchy throat; Skin symptoms: dry, itchy and reddened skin; Neurotoxic symptoms: headache, lethargy, and difficulty concentrating; Nonspecific symptoms: nasal congestion, runny nose, and wheezing.

The questionnaire results showed that 86% employee had the above-mentioned symptoms. Only 12% of those who had the above symptoms after improvement.

  • As commented previously, what is the justification of the locations of the sensors adjacent to the printer? The equipment and loggers should be close to the sitting area of a person to give a reliable value of a breathable area.

ANS: Thank you for your kind suggestion. We added the floor plan to mark the sensor locations. Please refer to this revised paper.

  • The hours to report the air temperature and relative humidity should be related to working hours or the tine HVAC system is on or off. At this point, the compression of these two variables does not give any significant findings.

ANS: Thank you for your kind assistance. We did rewrite the analysis section of this revised paper, focused on the working hours. Please refer to this revised paper.

  • The symbols in Table 2 should be defined. In table 2, on some days there is no significant difference between pre and after tests but some days shows higher differences. This inconsistent data makes the findings to be unreliable. Why is this happening?

ANS: Thank you for your kind suggestion. It is very useful for us to modify the presentation of this paper. We did rewrite the Comprehensive analysis and discussions and conclusions. Please refer to this revised paper.

  • What kind of statistical test is used to compare the data?

ANS: Thank you for your kind suggestion. The test results before and after changes were used the statistical method to compare the improving efficiency of the most economical way. Please refer to the analysis results of this revised paper.

  • The study lacks strong discussion to compare the findings of this study with other studies. The authors should read similar articles and compare the findings with the current study.

ANS: Thank you for your kind suggestion. It is very useful for us to modify the presentation of this paper. We tried to compare with the other research findings. Most of those results would took a lot of money to modify the indoor temperature, relative humidity and indoor air quality such as: added more air-conditioners, external sunshade systems and water spray systems and so on. Therefore, we did rewrite the Comprehensive analysis and discussions and conclusions. Please refer to this revised paper.

  • The contribution of the study should be highlighted in the discussion which is missing.

ANS: Thank you for your kind suggestion. It is very useful for us to modify the presentation of this paper. We did rewrite the Comprehensive analysis and discussions and conclusions. Please refer to this revised paper.

  • Part of the conclusion is not supported by the data regarding the temperature and humidity to be improved “greatly”. There is no justification for this claim. Lower or higher temperature is not an indicator of an improved temperature but the subject's report can be in the index if there is a statistical comparison between occupants' feedback to different temperatures.

ANS: Thank you for your kind suggestion. We did rewrite the conclusion section to highlight the finding of this paper. Please refer to this revised paper.

These comments are very useful for us to modify this paper. We are deeply appreciated for your kind suggestion and assistance. Please refer to this revised paper and review this paper. If there is any question, please feel free to contact me directly. I am looking forward to hearing from you soon. 

Best regards,

 Wen-Pei Sung

Reviewer 2 Report

  1. English must be improved, some sentences look like they have been translated by google translate, they make sense, but you would not word them in such a way in English. Ex line 69: The improving strategies with the most economical way were proposed to improve the indoor environment in this study based on the characteristics of the study site. Also, many repetitions of one word in a sentence.  Line 85: do not start a sentence with “actually”, you can just start with “when” and in general line 85 is very poorly written.
  1. Abstract, do not use abbreviations in the abstract (MFD, TSP). Introduce them the first time they occur in the text after the abstract. This is a common rule for abstracts.
  2. Line 28: “Especially, carbon dioxide and aerosols in the air were identified as key factors in office staff satisfaction” – add citation eg:
  • Schnotale J. “ CFD simulations and measurements of carbon dioxide transport in a passive house” Refrigeration Science and Technology, Pages 4065 – 4072, 2015, 24th IIR International Congress of Refrigeration, ICR 2015Yokohama
  • Szczepanik “Improving Household Safety via a Dynamic Air Terminal Device in Order to Decrease Carbon Monoxide Migration from a Gas Furnace” International Journal of Environmental Research and Public Health,Volume 19, Issue 3February-1 2022
  1. Line 34- your references are pointing to the same contaminants and the same type of spaces additional references with a different approach can be used here also to widen the range of application e.g. DOI: 10.1051/E3SCONF/20184400172
  2. Lines 40-41 – add citation
  3. Line 49: “The accumulation of indoor air pollutants seems to be closely related to sick building syndrome” – add citation/s
  4. Add to the introduction what you mean by “before and after improvement” – you use this a lot and it's not clear what you mean and what exactly you are improving until later in the text. This should be in the introduction

Authors should put: After on-site investigation and considering financial issues, the most economical improved plans, (1) add a mandatory ventilation system; (2) restore the air exchange system; (3) clean the air conditioning filter; (4) add ceiling openings to set up multiple air-back filters; (5) Replace the impervious office grid; (6) replace heavy carpet – also in the introduction or at the very beginning of the methods section.

  1. In Figure 1: mark the building with a red circle or something like this
  2. I would change the order in the methods section – first I would describe the building and location, what and how is measured and then the different mathematical indicators that you used. This would make the article more structured and easier to follow by the reader.
  3. In figure 2 – instead of 2 photos of the building, add the floor plan with dimensions of the test area so the reader can see what you are talking about. You can also put the “point A and B” locations of the sensors. From the description of sensor locations in lines 174 to 179 I have no idea where the sensors are. Without this, the study is very unclear
  4. Combine figures 3 and 4 into one or delete them both – they are less needed
  5. Line 206: dot use “27 degrees C” but 27°C – the same to all the other lines like this
  6. Line 210: do not use military time “18 o'clock”, use 6pm – the same to all the other lines like this
  7. Figure 5 is unclear – put everything on one graph or make them bigger and make sure the text on one axis does not cover the other axis like it is now.
  8. In Tables 2 and 3: change the names of “Pretest” and “Posttest” to something that is not so similar, the latter can be “after changes” or something like this. The way they are now is too similar and the tables are unclear. The word “Posttesd” does not exist in the English language and especially it is not a professional word for a scientific article.
  9. Line 277: World Health Organization, WHO standard. Otherwise, the air quality of PM2.5 concentration should be less than 12 μg/m3- add citation/s
  10. Why are the graphs in figure 6 from a different program than the ones in figure 5? If possible, please make them similar. If they are from different software please try to unify. Additionally, in the current form, fig. 6 is out of focus and stretched in the vertical direction so the text is stretched as well and looks unprofessional for a scientific paper.
  11. Figure 8 – please have the same scale for both figures, now it looks like there was no improvement
  12. The “Comprehensive analysis and discussions” chapter needs to be redone. It is very unclear.

Conclusions:

Although the topic of the article is interesting the lack of good structure, English language, lack of sufficient links to literature makes it a borderline article to be rejected. Extensive changes and editing are required for the article to make it sufficient as a scientific paper.

Author Response

Dear Sir/ Madam:

The answer for point to point of comments of Manuscript ID buildings-1625161 The Strategy for Improving Indoor Environment of Office Space in subtropical City has been modified following the reviewers’ comment. is as follows. Please check it again and see if it is acceptable for publication on Buildings.

Comments and Suggestions for Authors

  1. English must be improved, some sentences look like they have been translated by google translate, they make sense, but you would not word them in such a way in English. Ex line 69: The improving strategies with the most economical way were proposed to improve the indoor environment in this study based on the characteristics of the study site. Also, many repetitions of one word in a sentence.  Line 85: do not start a sentence with “actually”, you can just start with “when” and in general line 85 is very poorly written.

ANS: The English presentation of this paper has been modified by a Native English speaker-Dr. John Ring, Faculty of National Normal University, Taiwan. Please refer to this revised paper.

  1. Abstract, do not use abbreviations in the abstract (MFD, TSP). Introduce them the first time they occur in the text after the abstract. This is a common rule for abstracts.

ANS: Thank you for your kind suggestion. Abbreviations of MFD and TSP have removed. Please refer to this revised paper.  

  1. Line 28: “Especially, carbon dioxide and aerosols in the air were identified as key factors in office staff satisfaction” – add citation eg:

ANS: Thank you for your kind suggestion. We added the references such as :

Schnotale J. “ CFD simulations and measurements of carbon dioxide transport in a passive house” Refrigeration Science and Technology, Pages 4065 – 4072, 2015, 24th IIR International Congress of Refrigeration, ICR 2015Yokohama

Solange Leder, Guy R. Newsham, Jennifer A. Veitch, Sandra Mancini & Kate E. Charles (2016) Effects of office environment on employee satisfaction: a new analysis, Building Research & Information, 44:1, 34-50, DOI: 10.1080/09613218.2014.1003176

Szczepanik “Improving Household Safety via a Dynamic Air Terminal Device in Order to Decrease Carbon Monoxide Migration from a Gas Furnace” International Journal of Environmental Research and Public Health,Volume 19, Issue 3February-1 2022

  1. Line 34- your references are pointing to the same contaminants and the same type of spaces additional references with a different approach can be used here also to widen the range of application e.g.

ANS: Additionally, Air quality of office space would be deteriorated according to the poor ventilation of the parking lot with its high airtightness. This reference has been added in the introduction. Please refer to this revised paper.

Nina Szczepanik-Ścisło and Łukasz Ścisło, Air leakage modelling and its influence on the air quality inside a garage, E3S Web of Conferences 44, 00172 (2018) DOI: 10.1051/E3SCONF/20184400172

  1. Lines 40-41 – add citation

ANS: U. S. Environmental Protection Agency, Progress Cleaning the Air and Improving People's Health, https://www.epa.gov/clean-air-act-overview/progress-cleaning-air-and-improving-peoples-health

  1. Line 49: “The accumulation of indoor air pollutants seems to be closely related to sick building syndrome” – add citation/s  

ANS: We added the reference. Please refer to this revised paper.

  1. Norhidayah, Lee Chia-Kuang, M.K. Azhar, S. Nurulwahida, Indoor Air Quality and Sick Building Syndrome in Three Selected Buildings, Procedia Engineering, 2013, 53(2013), 93-98, DOI:10.1016/j.proeng.
  1. Add to the introduction what you mean by “before and after improvement” – you use this a lot and it's not clear what you mean and what exactly you are improving until later in the text. This should be in the introduction

Authors should put: After on-site investigation and considering financial issues, the most economical improved plans, (1) add a mandatory ventilation system; (2) restore the air exchange system; (3) clean the air conditioning filter; (4) add ceiling openings to set up multiple air-back filters; (5) Replace the impervious office grid; (6) replace heavy carpet – also in the introduction or at the very beginning of the methods section.

ANS: Thank you for your kind suggestion. We have already put the description of these strategies in the introduction section. Please refer to this revised paper.

  1. In Figure 1: mark the building with a red circle or something like this

ANS: Thank you very much for your kind suggestion. We added a red circle on the Fig. 1. Please refer to this revised paper.

  1. I would change the order in the methods section – first I would describe the building and location, what and how is measured and then the different mathematical indicators that you used. This would make the article more structured and easier to follow by the reader.

ANS: Thank you for your kind suggestion. We did change these two sections. Please refer to this revised paper. 

  1. In figure 2 – instead of 2 photos of the building, add the floor plan with dimensions of the test area so the reader can see what you are talking about. You can also put the “point A and B” locations of the sensors. From the description of sensor locations in lines 174 to 179 I have no idea where the sensors are. Without this, the study is very unclear

ANS: Thank you for your kind suggestion. We added the floor plan to mark the sensor locations. Please refer to this revised paper.

  1. Combine figures 3 and 4 into one or delete them both – they are less needed

ANS: Thank you for your kind assistance. We deleted Fig. 3 and added the floor plan of this test.

  1. Line 206: dot use “27 degrees C” but 27°C – the same to all the other lines like this

ANS: We have already modified all of these presentations. Thank you a lot.

  1. Line 210: do not use military time “18 o'clock”, use 6pm – the same to all the other lines like this

ANS: Thank you very much. We did changes all of them.

  1. Figure 5 is unclear – put everything on one graph or make them bigger and make sure the text on one axis does not cover the other axis like it is now.

ANS: Thank you very much. Figure 5 has been modified. Please refer to this figure.

  1. In Tables 2 and 3: change the names of “Pretest” and “Posttest” to something that is not so similar, the latter can be “after changes” or something like this. The way they are now is too similar and the tables are unclear. The word “Posttesd” does not exist in the English language and especially it is not a professional word for a scientific article.

ANS: Thank you very much. We changed them to “ Before changes” and “After changes”.

  1. Line 277: World Health Organization, WHO standard. Otherwise, the air quality of PM2.5 concentration should be less than 12 μg/m3- add citation/s

ANS: Thank you for your kind suggestion. We did add the reference: U. S. Environmental Protection Agency, The National Ambient Air Quality Standards for Particle Pollution, Revised Air Quality Standards for Particle Pollution and Updates to the Air Quality Index (AQI), 2012  https://www3.epa.gov/airquality//

Environmental Protection Administration, Executive Yuan, Taiwan, Taiwan Air Quality Index, AQI, https://airtw.epa.gov.tw/cht/Information/Standard/AirQualityIndicator.aspx

  1. Why are the graphs in figure 6 from a different program than the ones in figure 5? If possible, please make them similar. If they are from different software please try to unify. Additionally, in the current form, fig. 6 is out of focus and stretched in the vertical direction so the text is stretched as well and looks unprofessional for a scientific paper.

ANS: Thank you very much for your kind suggestion. The presentation of Figure 5 and 6 have been modified. We have already used the same scale to compare the improved efficiency before and after changes.  

  1. Figure 8 – please have the same scale for both figures, now it looks like there was no improvement

ANS: Thank you very much for your kind suggestion. We have already used the same scale to compare the improved efficiency before and after changes. 

  1. The “Comprehensive analysis and discussions” chapter needs to be redone. It is very unclear.

ANS: Thank you very much for this kind suggestion. We did rewrite this section. Please refer to this revised paper.

Conclusions:

Although the topic of the article is interesting the lack of good structure, English language, lack of sufficient links to literature makes it a borderline article to be rejected. Extensive changes and editing are required for the article to make it sufficient as a scientific paper.

ANS: Thank you for your kind suggestion. We did modify this paper a lot. Please refer to this revised version of this paper.

These comments are very useful for us to modify this paper. We are deeply appreciated for your kind suggestion and assistance. Please refer to this revised paper and review this paper. If there is any question, please feel free to contact me directly. I am looking forward to hearing from you soon. 

Best regards,

Wen-Pei Sung

Round 2

Reviewer 1 Report

The authors have improved the manuscript significantly. The quality and scientific soundness has improved.

I have no further comments and the manuscript is ready for publication.

Author Response

Dear Sir/ Madam:

The answer for point to point of comments of Manuscript ID buildings-1625161 Strategy for Improving the Indoor Environment of Office Spaces in Subtropical Cities has been modified following the reviewers’ comment is as follows. Please check it again and see if it is acceptable for publication on Buildings.

Comments and Suggestions for Authors

The authors have improved the manuscript significantly. The quality and scientific soundness has improved.

I have no further comments and the manuscript is ready for publication.

ANS: Thank you for your kind comments. We are very appreciated for your kind suggestions for us to modify this paper.  

All authors are deeply appreciated for your kind suggestion and assistance. Please refer to this revised paper and review this paper. If there is any question, please feel free to contact me directly. I am looking forward to hearing from you soon. 

Best regards,

 Wen-Pei Sung

Reviewer 2 Report

Dear Authors,

Thank you for submitting the revised manuscript. The improvement is evident, however, there are still some minor elements to be addressed.

A) General remarks

  1. The paper is much clearer now. In the current version, the literature in the paper is adequately cited with a sufficient number of references. The English and some elements of the not proper way of writing the scientific paper were improved. Both the abstract and the conclusions were improved and the reviewer has no other remarks on those parts.
  2. The introduction is also improved
  3. All the figures mentioned in the reviewers' first stage review were corrected. Some additional comments:

- Fig.1 the pointer (red almost circle) on the map is evidently made in “paint” of other similar software it looks very unprofessional.

- Fig. 3 is good and clarifies a lot however the text is often crossing the lines, seems to be of different sizes, the arrow is a different weight. Thus it requires editing to look professional. Additionally, the caption is suggested to be changed- not “this test office” but “the test office”.

  1. Additionally, the article still requires some editing e.g.:

- line 143 and following units μ m with extra spaces,

- lines 191-195, 220-223 some symbols are inserted from the symbol list and some others as text thus they look like written in different styles/sizes. Consider placing all in the same convention (e.g. like quotations) so they look the same and professional.

  1. Avoid using not definite statements like “can” “may”. The authors are using it a lot but in the scientific paper, only definite statements are to be used. This must be improved.
  2. It looks that in many places the author change the names of the data before and after improvements (as suggested in the first stage review). However, in many places, the old way of colling this “pretest” “posttest” is still present. Now with both conventions, it is really difficult to connect data to the explanations. This is present especially in:

- fig.5, although the caption has changed the caption above the graphs, is still with the old name. The same for Fig.6, 7,8. Also  9 and 10(legend).

-table 2 and 3, 4

Conclusions:

The article after minor changes can be considered for publication.  However, it must be pointed out that the weakest element of the article is not paying attention to details, not fully professional figures and tables. This seems that the article is rushed. For future reference, authors must focus more on the way they present their results.

Author Response

Dear Sir/ Madam:

The answer for point to point of comments of Manuscript ID buildings-1625161 Strategy for Improving the Indoor Environment of Office Spaces in Subtropical Cities has been modified following the reviewers’ comment is as follows. Please check it again and see if it is acceptable for publication on Buildings.  

Comments and Suggestions for Authors

Dear Authors,

Thank you for submitting the revised manuscript. The improvement is evident, however, there are still some minor elements to be addressed.

  1. A) General remarks
  1. The paper is much clearer now. In the current version, the literature in the paper is adequately cited with a sufficient number of references. The English and some elements of the not proper way of writing the scientific paper were improved. Both the abstract and the conclusions were improved and the reviewer has no other remarks on those parts.

ANS: All authors are very appreciated for your kind suggestions to modify the presentation of this paper.

  1. The introduction is also improved

ANS: All authors are very appreciated for your kind suggestions to modify the presentation of this paper.

  1. All the figures mentioned in the reviewers' first stage review were corrected. Some additional comments:

- Fig.1 the pointer (red almost circle) on the map is evidently made in “paint” of other similar software it looks very unprofessional.

ANS: Thank you for your kind suggestion. This figure has been modified. Please refer to this revised paper.

- Fig. 3 is good and clarifies a lot however the text is often crossing the lines, seems to be of different sizes, the arrow is a different weight. Thus it requires editing to look professional. Additionally, the caption is suggested to be changed- not “this test office” but “the test office”.

ANS: Thank you for your kind suggestion. This figure and caption have been modified. Please refer to this revised paper.

  1. Additionally, the article still requires some editing e.g.:

- line 143 and following units μ m with extra spaces,

- lines 191-195, 220-223 some symbols are inserted from the symbol list and some others as text thus they look like written in different styles/sizes. Consider placing all in the same convention (e.g. like quotations) so they look the same and professional.

ANS: Thank you for your kind suggestion. The presentation of this part has been modified. Please refer to this revised paper.

  1. Avoid using not definite statements like “can” “may”. The authors are using it a lot but in the scientific paper, only definite statements are to be used. This must be improved.

ANS: Thank you for your kind suggestion. We did modify all statements of “can” “may”.  Please refer to this revised paper.

  1. It looks that in many places the author change the names of the data before and after improvements (as suggested in the first stage review). However, in many places, the old way of colling this “pretest” “posttest” is still present. Now with both conventions, it is really difficult to connect data to the explanations. This is present especially in:

- fig.5, although the caption has changed the caption above the graphs, is still with the old name. The same for Fig.6, 7,8. Also  9 and 10(legend).

-table 2 and 3, 4

ANS: Thank you for your kind suggestion. We did modify “pretest” “posttest” as “Before Change” “After Change” respectively. The captions of Fig. 6~10 and Table 2~4 have been modified. Please refer to this revised paper.

Conclusions:

The article after minor changes can be considered for publication.  However, it must be pointed out that the weakest element of the article is not paying attention to details, not fully professional figures and tables. This seems that the article is rushed. For future reference, authors must focus more on the way they present their results.

We are deeply appreciated for your kind suggestion and assistance. These comments are very useful for us to modify this paper. Please refer to this revised paper and review this paper. If there is any question, please feel free to contact me directly. I am looking forward to hearing from you soon. 

Best regards,

Wen-Pei Sung
